# Improvement of Packaging Circularity through the Application of Reusable Beverage Cup Reuse Models at Outdoor Festivals and Events

**Valdonė Šuškevičė** *[ID] and **Jolita Kruopienė** [ID]

Institute of Environmental Engineering, Kaunas University of Technology, LT-44249 Kaunas, Lithuania; jolita.kruopiene@ktu.lt
* Correspondence: valdone.daugelaite@ktu.lt; Tel.: +370-615-95-760

**Abstract:** Festivals generate huge amounts of waste during a short period of time, usually in three to four days. Single-use packaging is one of the dominant waste streams at the festivals. In order to minimize single-use plastic packaging waste generation and negative impacts on the environment, outdoor festivals apply alternative reusable cup systems and strategies. However, little studies have been made on how different reusable beverage cup reuse models can affect material circularity within certain festivals, and how it contributes to cup damage and loss. This article presents the results of a pilot study of different reusable cup reuse models within seven Lithuanian summer outdoor festivals. Three different models were applied and tested: A—only reusable cups, non-refundable model; B—only reusable cups, with deposit-refund; C—a mixed system of reusable cups with deposit-refund and of single-use cups. Material flow analysis (MFA) was performed, and the Materials Circularity Indicator (MCI), developed by Ellen MacArthur Foundation, was calculated to study the applied models. According to the findings, refund models (B, C models) have lower rates of damaged and lost cups compared to non-refundable reusable cup reuse model (A model). This paper shows that different reuse models provide different damage, loss and return rates of reusable cups. The data presented can aid decision-makers who need to choose a reuse model for a certain event.

**Keywords:** circular economy; reusable cups; sustainable events; deposit-refund system; packaging waste; reuse systems; material flow analysis (MFA); circularity indicators; waste prevention; resource efficiency



## 1. Introduction

Global plastics consumption rates have been increasing steadily since the 1950s [1]. In 2015, global plastics production had reached 380 million metric tons with around 40% used for packaging [2]. Packaging is recognized as the dominant type of single-use plastic in the market with around 60% of all plastic packaging used for the food and beverage market [3]. Further still, the consumption of plastic packaging is on the rise [2].

Geyer et al. calculated that, in 2015, around 6300 MT of plastic waste had been generated, from which only 9% was recycled, 12% incinerated, and 79% was accumulated in landfills or the natural environment [4]. These authors presumed that this linear waste management trend can lead to roughly 12,000 MT of plastic waste entering the natural environment and landfills by the year 2050.

Single-use plastics have potential to cause a negative impact on the environment through littering and accumulation, as well as landfilling and incineration. Inappropriate disposal, inadequate management of plastics and the linear single-use plastic consumption model can become a source of leakage to the natural environment leading to pollution in many forms [2], and negative environmental effects on natural habitats [1,5]. Every year from 5 to 13 million tons of plastic are estimated to end up in the ocean, and plastic pollution reaches even the most remote areas of the planet [6]. Single-use plastics make

up 86% of all single-use items found on beaches. Beverage cups are one of the top ten single-use items found there [7]. As single-use plastics are designed to be used once, this leads to the loss of this valuable material, which plays an important role in our economy.

To reduce environmental damage, efforts are now needed to significantly increase the recycling rates of packaging plastics [8]. The latest report by the European Court of Auditors reveals that new calculation methods for the actual share of plastic packaging being recycled could drop the value down from 42% to 30% across EU member states [9]. In some cases, waste collection is missing or is difficult to implement, and consequently, waste recycling performances are low; this is the case of the festivals. Industry research showed that only 32% of single-use PET (polyethylene terephthalate) cups used in UK festivals are recycled [10].

The European Commission has reviewed Directive 94/62/EC to reinforce the mandatory essential requirements for packaging to be allowed on the EU market with a focus on reduction of packaging and packaging waste through waste prevention measures and design for reuse and recyclability of packaging [11]. A new Circular Economy Action Plan, represented by the European Commission in 2020, also aims to accelerate the transition from materials recycling to waste prevention and reuse strategies [12,13]. The Plan gives incentives that pursue the transformation of current packaging systems to more circular and sustainable ones [12,14]. The Single-Use Plastics Directive that entered into force on 2 July 2019 is another significant step forward to urge the transition from single-use plastics towards reusable products and systems [15]. The directive foresees certain policy measures, such as market restriction, new product design requirements, extended producer responsibility (EPR) schemes, awareness-raising measures, consumption reduction, improved collection and labeling requirements as the key elements to move away from single-use plastic products [8,16].

Relatively limited use and application of reusable packaging reuse systems today might be the result of the global drift from reusable packaging to single-use packaging during the past decades [17]. Nevertheless, Coelho et al. (2020) claim that packaging reuse is not new in both B2B (business to business), and B2C (business to consumer) segments, where the application of different forms of reusable packaging solutions have been established for primary and secondary, to tertiary (transportation) packaging [18]. Mahmoudi and Parviziomran (2020) point out that even reuse practices are not new, but primary packaging, that has direct contact with the product, is recognized to be a newer concept in comparison to secondary and tertiary packaging reusability. Additionally, changes in packaging ownership have been observed—the authors state that there is a clear move from packaging owning to renting services [18]. This could have a significant impact on our consumption patterns. Coelho et al. (2020) see the growth of acceptance for service systems, instead of ownership in the reusable packaging market. This ownership transformation has both some evidential advantages and challenges for the decision-makers. As the authors have noted, reusable packaging return systems may increase customers' loyalty to retailers through refund schemes. The authors identify advantages for the reusable returnable packaging consumers, which are cost reduction through discounts, as well as the reduction in waste amounts [17].

Economic incentives can play a big part in reducing the consumption of single-use plastic [15]. There is strong evidence that charges have a better performance than discounts in reducing the use of single-use disposable beverage cups. A review carried out by the University of Cardiff revealed that a minimum charge of £0.20 would be needed to change the behavior of 49% of the population [19]. Moreover, social marketing tools and information campaigns can contribute to the success of the charges [19]. Increased and improved information and raising consumer awareness go hand-in-hand with the regulatory measures to reduce consumption, redesign products and contribute to ending plastic pollution [15].

Festivals and events in a short period of time (from a few hours to a few days) generate huge amounts of single-use packaging that become one of the dominant waste streams at

these events due to a large number of people, intensive activities, and location characteristics [20,21]. The waste hierarchy concept shows that waste prevention and minimization is the most efficient and effective means of resource management. Therefore, different waste prevention measures are being implemented at different festivals worldwide to tackle the problem of single-use packaging, beverage cups included. The most popular waste prevention measures applied by green-minded festivals are reusable packaging deposit-refund models, package-free alternatives, festival rules for the participants to carry their own dishes, and special requirements (guidelines) for the vendors to provide certain types of packaging or alternative packaging systems at the festival venue [22].

In 2017, Linder et al. presented three levels of circularity where different activities can be applied in order to empower the transition to circular economy: micro level (individual companies and consumers), meso level (eco-industrial parks), and macro level (city, province, region, and nation) [23]. Festivals can be seen as micro-level closed systems, where material inputs and outputs can be measured and easily controlled. Maintaining the value of products, materials, and resources for as long as possible is the fundamental goal of circular economy, and the reuse of packaging can help achieve this. It is recognized that deposit-refund schemes help to manage and control resource recovery from the consumers.

The majority of authors agree that reuse of packaging has a huge potential for environmental impact reduction through the decreased need of materials, especially virgin or primary materials, when the reuse systems are well developed and function sustainably [17,24].

A comprehensive comparative life cycle analysis of different cup systems for selling of drinks at events performed by Pladerer et al. (2020) revealed that reusable cup models have lower environmental burden compared to disposable cup models, despite the fact that life cycle assessments (LCA) for reusable cups were always based on the most disadvantageous scenario compared to disposable cups [25]. According to the Realising Another World (RAW) Foundation [26], a reusable cup is the better environmental option compared to a disposable polyethylene terephthalate (PET) cup, after more than 3 uses in many cases, and after around 9–14 uses compared to cardboard. Another analysis showed that after 10 uses of reusable polypropylene (PP) cup, it generates less environmental impact compared to single-use polypropylene cups [27].

Nevertheless, reuse packaging systems may have a higher environmental footprint compared to single-use packaging if the reuse systems are not being managed carefully in a sustainable way. Most of the negative impacts of reusable packaging reuse systems have the potential to occur through certain aspects such as inappropriate packaging design, complex logistics, containers' service life, cleaning, food safety, etc. [17,18]. Coelho et al. (2020) agree that economics, as well as environmental impacts of reuse systems, strongly depend on several factors like logistics, reusable packaging return rates, cleaning, labor costs, and market size. The authors note that more studies of life cycle costing have to be made to compare the costs of single-use and reusable packaging systems.

A total of 29 festivals and festival-like events were organized during the 2019 summer season in Lithuania where standard linear "take-make-dispose" model of disposable single-use beverage cups was applied. No clear measures have been made in order to minimize the use of disposable beverage cups and reduce the waste of single-use plastics within the events. Until then, event organizers had no alternative to disposable beverage cups. Several events used reusable cup deposit systems in the past, but the cups for these events had promotional attributes and were therefore used as souvenirs. Single-use plastic beverage cups remain the most popular packaging choice for cold drinks like beer, cider, and cocktails in Lithuanian open-air festivals. However, such a choice significantly increases waste treatment costs for the festival organizers due to a big volume of waste, additional labor, and time needed for the clean-up of the area after the festival. A more resource-efficient and circular approach needs to be realized.

The objective of our work was to assess the environmental benefits of cup reuse models in open summer festivals on material circularity and the avoidance of disposable

plastic. The paper describes the performance of service systems implemented in Lithuanian festivals with the aim to identify which models (and market-based instruments) can optimize the use of reusable cups and minimize losses and damages. The aim of this work was achieved by answering three research questions (RQs):

1.  How many cups will be used, lost, and damaged during the festivals when different reusable cup reuse models are applied (RQ1)?
2.  What impact do reusable cup reuse models make on material circularity compared to the use of single-use beverage cups at the festivals (RQ2)?
3.  How many single-use plastic cups were avoided by applying reusable beverage cup reuse models at the festivals (RQ3)?

## 2. Materials and Methods

Three alternative reusable beverage cup reuse models were implemented as pilot tests at 7 Lithuanian open-air festivals during the summer season of 2019 in order to eliminate single-use plastic cups. "Take-make-use-reuse" model was provided by the so-called "CupCup" initiative for each festival instead of a single-use "take-make-use-dispose" model.

Festivals varied in size, style, attendees' behavior, and the implemented reuse model. Those models were (Figure 1):

Model A: only reusable cups, non-refundable;
Model B: only reusable cups with deposit-refund;
Model C: a mixed system of reusable cups with deposit-refund, and of single-use cups.

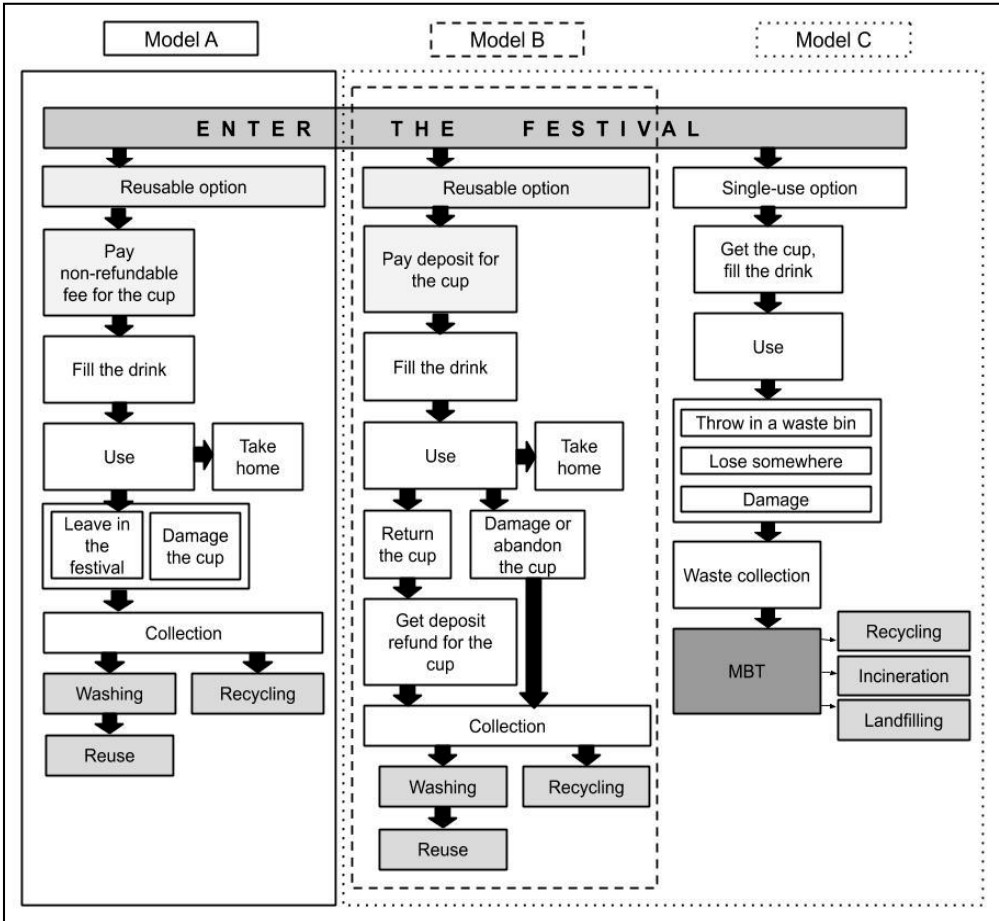

**Figure 1.** Three tested reuse models of reusable cups at seven Lithuanian open-air festivals. MBT— mechanical biological treatment plant.

### 2.1. Case Studies

Case study was chosen as a research method in order to represent the data of real-life situations and to provide better insights into the detailed behaviors of the subjects of interest [28]. Seven micro-level events, namely open-air festivals, were chosen. These were the festivals whose organizers agreed to test the practice of using reusable cups. The events took from 2 to 4 days, had from 500 to 6000 attendees, and applied different reusable cup reuse models (Table 1). Participants of the events had the option to exchange dirty cups for clean cups an unlimited number of times. One-size (500 mL) reusable plastic cups weighing 32 g per unit, made from polypropylene (PP, no. 5) were used. Dirty cups were washed at the washing center outside the festivals. A simplified scheme of reusable and single-use cup flows is presented in Figure 2.

**Table 1.** Summary of the festivals, the applied reuse models, and additional characteristics of each festival.

| Festival | 1 | 2 | 3 | 4 | 5 | 6 | 7 |
|---|---|---|---|---|---|---|---|
| Number of attendees | 1500 | 900 | 3000 | 500 | 950 | 700 | 2000 |
| Duration, days | 3 | 3 | 4 | 2 | 3 | 2 | 3 |
| The applied reuse model for beverage cups | A1 | A2 | B1 | B2 | B2 | B2 | C |
| Festival characteristics (time, location) | July 4–7, Palanga | August 8–11, Trakai district | August 28–September 1, Nida | June 14–16, Vilnius region | July 25–28, Utena region | August 9–11, Trakai district | July 11–14, Anykščiai |
| Average age of attendees (years) | 30+ | 30+ | 35+ | 20+ | 20+ | 25+ | 25+ |
| Main concept, activities | Special guests, high standards for attendance: only with costumes, higher price tickets, secret and very secure festival party. | Mature attendees; over 50% foreigners; most of them artists, hipsters, creative people. The festival had strict entrance and ticketing policy—organizers chose to whom they sell the tickets. | Open-air event at the city center, free of charge, no restricted territory or ticketing. Live music, food, drinks, yacht racing. | Small punk festival, alternative, experimental music. The main concept of the festival was a green and sustainable event. Attendees are younger age, lower entrance fee. | Electronic music festival. Strict face control, people are long-term participants, higher price tickets, far away from city, in the middle of woods. | Electronic music festival, dark music, post punk, lot of DJ sets, some experimental music, people are long-term participants, far away from city, surrounded by nature. | Metal, hard rock music, niche festival, international, popular among Baltic states' hard rock fans. Has its own audience, most attendees are fans of this festival and attend it many years as a tradition. |

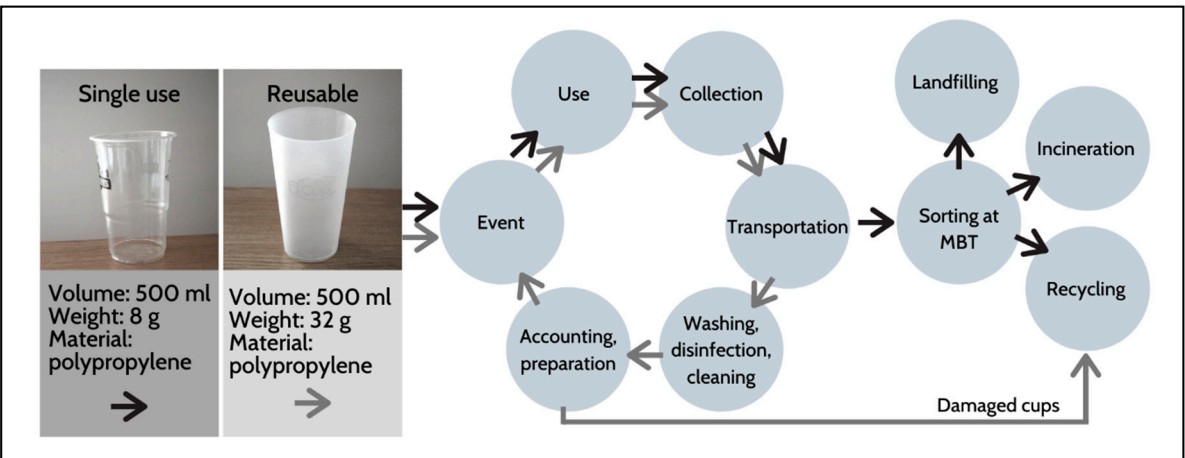

**Figure 2.** A simplified scheme of reusable and single-use cup flows.

### 2.1.1. Model A: Only Reusable Cups, Non-Refundable

Some of the festivals preferred a non-refundable reuse model in order to simplify processes and money flows. Two types of non-refundable models, A1 and A2, were applied:

A1: Festival took a non-refundable eco-fee, and organizers collected the cups during the whole event from the territory; no penalty for lost cups was applied;

A2: Festival took a non-refundable eco-fee, but if the cup was lost during the event, a new cup cost 1 euro extra fee.

### 2.1.2. Model B: Only Reusable Cups with Deposit-Refund

This model provided partly and fully refundable reuse schemes for event attendees:

B1: Fully refundable reuse model. Full deposit was refunded for the attendees. This model is easy to apply and communicate, but the cost of the service has to be covered by organizers.

B2: Partly refundable reuse model. Only half of the amount was refunded when the cup was returned. The non-refundable part of the deposit was considered as a fee in order to cover service costs, and the other half was a refundable deposit that motivated attendees to return the cups. The B2 model was used for data verification. Three events applied the following reuse model, so this particular B2 model was chosen to verify the data, and check data reliability. Verification is used in order to identify how significant cup loss, damage, and return rates differ within three different events with the same reuse model (B2) applied.

### 2.1.3. Model C: A Mixed System of Reusable Cups with Deposit-Refund, and of Single-Use Cups

This model allowed the festival participants themselves to choose whether to use reusable or disposable cups. It was applied as a compromise due to a sponsorship agreement between the festival and drink producer, in order to retain an advertising position on single-use cups provided by drink producers. Organizers made an announcement before the event that there will be an alternative to disposable cups at the festival, and those who feel environmentally supportive will be able to use the reusable alternative. Partly refundable reuse scheme (as in model B2) was applied at this festival in order to cover service costs for providing reusable cups for vendors, and to allow customers to exchange used cups for clean cups an unlimited number of times throughout the whole event.

### 2.2. Material Flow Analysis (MFA)

Material Flow Analysis (MFA) has become a reliable instrument to describe material flows and stocks within various systems [29]. Substance Flow Analysis (STAN) software

was used to perform MFA according to the Austrian Standard ÖNORM S 2096. STAN is often used simply for displaying mass flows of goods and substances as Sankey arrows [29] (see Figures 1 and 2).

Material flow analysis (MFA) has a considerable history as an environmental accounting approach and in the field of industrial ecology. The flow of material resources, products, and wastes have been intensively studied during the past decade with the aim of characterizing the burden on the economy and setting targets for the sustainable use of material resources [30].

For this research, it was important to identify differences among three different reusable cup reuse models (A, B, C), and find out if there is a notable distinction among them in the context of reusable cup damage, loss, and return rates, as well as potential numbers of avoided single-use cups. Festivals are considered closed systems where certain activities and processes are being operated. Material flow analysis (MFA) was performed in order to measure reusable cup flow in and out of the system using Substance Flow Analysis software STAN [29].

### 2.3. Material Circularity Indicator (MCI)

Circularity metrics are useful for empirical assessment of the effects of a circular economy in terms of job creation, profitability, and environmental impacts [19]. The Ellen MacArthur Foundation with partners have created a Material Circularity Indicator evaluation methodology. It measures how restorative the material flows of a product or company are, and complementary indicators allow additional impacts and risks to be taken into account. The indicators can be used as decision-making tools for designers, but might also be used for several other purposes including internal reporting, procurement decisions and the evaluation or rating of companies. The MCI gives a value between 0 and 1 where higher values indicate a higher circularity. The MCI is calculated according to the following equation [31]:

$$MCIp = 1 - LFI \cdot F(X) \tag{1}$$

where LFI is the Linear Flow Index, and is associated with the linear product flow—sourced from virgin materials and ending up as unrecoverable waste. A product is called fully linear if it is made purely from virgin material and it completely goes into landfill or energy recovery after its use; that is, LFI = 1, which is calculated according to the following equation:

$$LFI = \frac{V + W}{2M} \tag{2}$$

In this case, the maximum value of 1 for LFI occurs when the mass of virgin raw material used in manufacture (V), and the mass of unrecoverable waste (W) are both equal to M—when there is no recycled (or reused) content and no collection for recycling (or reuse).

The minimum value for LFI occurs when V = W = 0, that is, when there is 100% recycled (or reused) content and 100% collection for recycling (or reuse).

The MCI depends on three main characteristics: V, W, and a utility X. In MCI calculations, the utility factor F(X) is depicted as a function of the utility X of a product. X shows the length and intensity of the product's use. These characteristics are expressed according to the following equations:

$$V = M \cdot (1 - F_R - F_U), \tag{3}$$

which reflects the material that is not from reuse, recycling or biological materials from sustained production. In this equation, M is the mass of a product; $F_R$ is the fraction of a product's feedstock mass that is taken from recycled sources; $F_U$ is the fraction of a product's feedstock mass that is taken from reused sources. The mass of unrecoverable waste (W) that is attributed to the product is expressed according to the following equation:

$$W = W_0 + \frac{W_F + W_C}{2}, \tag{4}$$

where $W_0$ is the mass of unrecoverable waste through a product's material going into landfill, waste to energy, and any other type of process where the materials are no longer recoverable; $W_F$ is the mass of unrecoverable waste generated when producing recycled feedstock for a product; and $W_C$ is the mass of unrecoverable waste generated in the process of recycling parts of a product. The utility X is expressed by the following equation:

$$X = \left(\frac{L}{L_{av}}\right) \cdot \left(\frac{U}{U_{av}}\right), \tag{5}$$

where L is the actual average lifetime of a product; $L_{av}$ is average lifetime of an industry-average product of the same type; U is the actual average number of functional units achieved during the use phase of a product; $U_{av}$ is the average number of functional units achieved during the use phase of an industry average product of the same type. The utility X has two components: one accounting for the length of the product's use phase (lifetime) and another for the intensity of use (functional units). Increasing the lifetime L when the industry average $L_{av}$ remains fixed leads to an increase in X and, correspondingly, to an increase (and thus an improvement) in the product's MCI.

## 3. Results and Discussion

Case studies at seven festivals have revealed the peculiarities, advantages and disadvantages of different reuse models for reusable cups.

### 3.1. Material Flow Analysis Results

Material flow analyses showed that each reuse model, and even smaller reuse categories (A1, A2, B1, B2, C) have relatively different cup loss, damage, and return for reuse rates, where certain patterns can be identified. B2 reuse model was chosen for data verification because it covered three different events.

Damaged cups were recycled to make pads for coffee cups in partnership with the Precious Plastic Lithuania initiative, and some of the cups with shape deformations and color changes were repurposed to souvenir pots for plants. Washing and cleaning of used cups took place at the washing center, where industrial washing equipment was used and detergents were applied for disinfection and proper cleaning of reusable cups. The loss was associated with taking cups home, or in some cases, with throwing them away with mixed waste, especially when no economic incentives for return were applied.

3.1.1. Material Flow Analysis of Different Reuse Models (A1, A2, B1, B2, C)

Reuse model A (both A1 and A2) had no economic incentives to return the cups after use; only a single-use fee was applied. The first type of event with reuse model A1 had no financial motivation for attendees to return and take care of the cups, and the second type of event with reuse model A2 had a small penalty for the loss of the cup, but no financial motivation for attendees to return the cups at the end of the festival. The results of reuse model A1 revealed a relatively high rate of total cup loss (22%) compared to the B and C models, but it was lower than for the A2 reuse model, for which the rate of total cup loss was 22.9% (see Figure 3) Reuse model A1 had the highest consumption rate of cups used for 100 participants per one festival day. Such a high consumption rate is associated with the model specifications like each drink being served in a new cup because most of the drinks were cocktails. Additionally, there was no need for taking care of your own cup—no penalties for lost cups were applied and most of the cups were left in the territory, at the bars and in special collection boxes that were spread throughout the territory. Reuse model A1 was the most intensive in consumption of beverages and was similar to single-use or disposable model with the principle "take-make-dispose (leave)". Nevertheless, several key factors led to the recovery of the cups: good cup collection from the territory by volunteers during the whole event; small, closed and well-defined festival area; large number of cup collection boxes spread through the territory; good communication with participants—the

organizers informed the participants about the system just before the event and showed an attractive video.

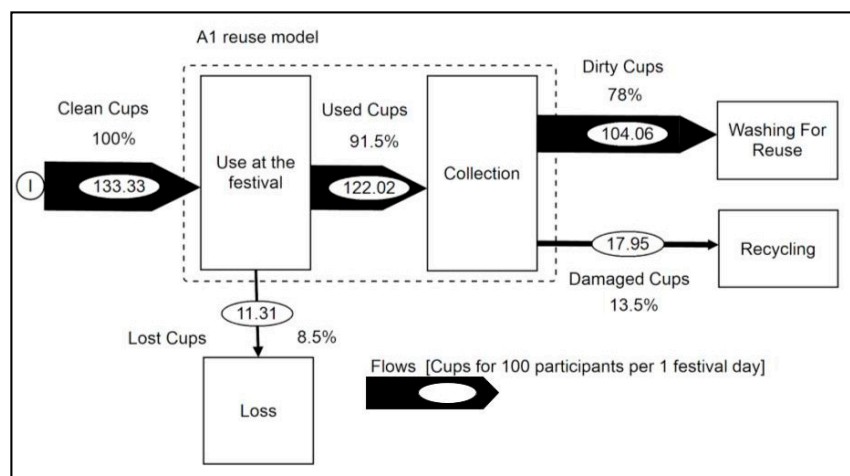

**Figure 3.** Material flow diagram of reusable cups at the festival where reuse model A1 was applied.

Reuse model A2 showed the highest rate of total cup loss—22.9% (see Figure 4). This can be explained by participants taking cups home as souvenirs, bearing in mind that more than half of the participants in this festival were foreigners. Cups lost in the festival territory were collected by the volunteers during the whole event. Most of the cup damage observed in this festival was aesthetic; that is to say, cups were colored, painted, or applied with stickers. However, in reuse model A1, most of the damage was crucial, consisting of cigarette burns and breaks, etc.

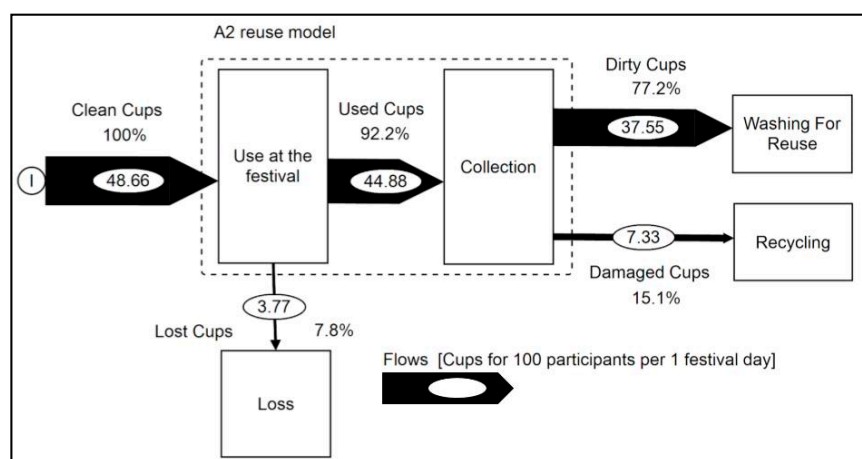

**Figure 4.** Material flow diagram of reusable cups at the festival where reuse model A2 was applied.

Reuse model B (both B1 and B2) had economic incentives to return the cups. This model motivated the participants not to break and not to lose the cups. If one participant lost their cup in the festival territory, someone else could return their cup to the eco-point or to the bar to get refunded.

The success of reuse model B1 is presumed to be due to the full refund of the deposit and to the dominant age group of the participants being over 30 years. Moreover, this festival was supported by the municipality, which insisted that organizers use reusable solutions instead of single-use plastics. The results show that the total loss of reusable cups was 3.1%, which is 11.9% lower than for reuse model B2 (see Figure 5).

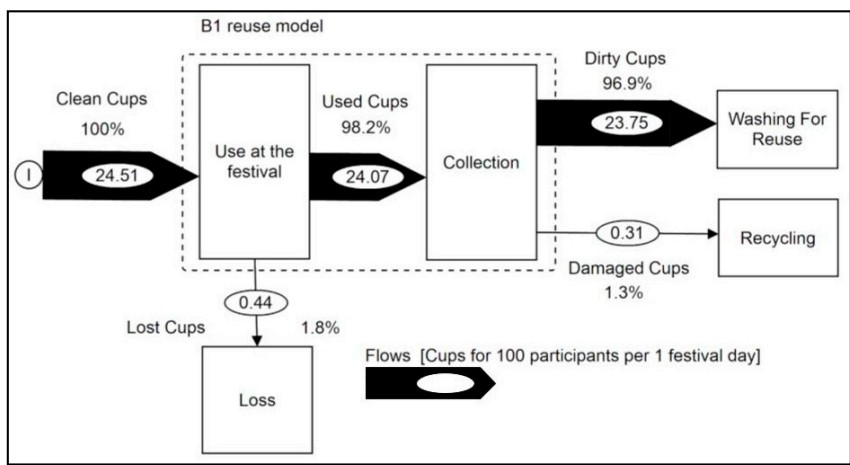

**Figure 5.** Material flow diagram of reusable cups at the festival where reuse model B1 was applied.

Reuse model B2 was partly refundable. Participants paid a refundable deposit, and half of the sum was refunded when the cup was returned without damage. This type of refund model is as effective as the fully refundable type of event. However, it is hard to apply this model for every event, especially for one-day events or city markets, where people are visiting episodically. It is also hard to apply when there are no defined boundaries of the event. In that case, the movement of people is chaotic, and the time spent at the event is relatively short—from 30 min to one day. This presumes that an extra fee for cup service would be an undesirable investment for the participant. In this case, event organizers would have to cover cup service fees, so the B1 refund model would have to be applied. Reuse model B2 was applied for three festivals, so this model was used for data verification. The average total loss of reusable cups in this model was 15% (see Figure 6).

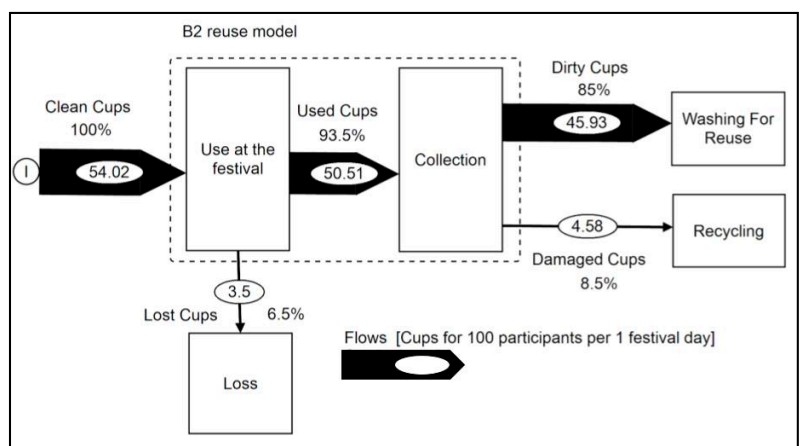

**Figure 6.** Material flow diagram of reusable cups at the festival where reuse model B2 was applied.

Reuse model C was partly refundable, as was reuse model B2. The only difference among these two models was that the B2 model had only a reusable solution for the participants, while the C model had both single-use and reusable choices. In this type of festival, participants were allowed to choose between single-use and reusable systems, so this model was chosen only by conscientious and motivated participants who felt responsible for the environment. For this reuse model, communication was a key success factor to reach more participants and increase the reuse rate. From 6000 participants, 2000 participants chose the reusable "CupCup" alternative, which accounts for one third of the total festival attendees. This reuse model had a relatively low total damage rate—2.5%, and total loss of reusable cups in this model was 6.6% (see Figure 7), which is associated with the consciousness and motivation of people who chose this alternative.

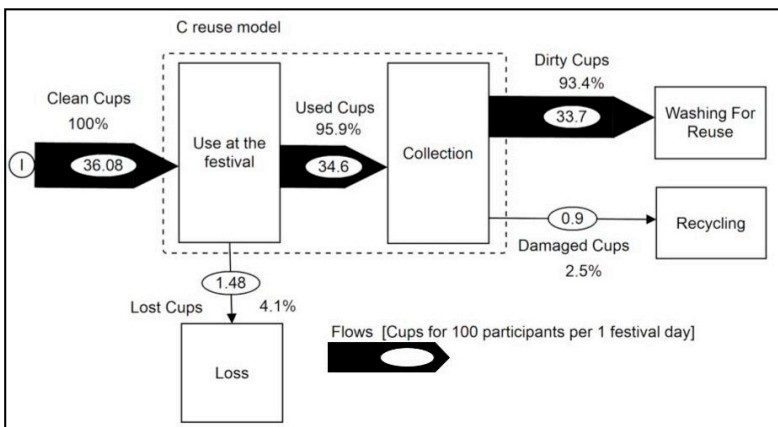

**Figure 7.** Material flow diagram of reusable cups at the festival where reuse model C was applied.

### 3.1.2. Cup Loss, Damage, and Return

Taken as a whole for all festivals, the total loss (damaged and lost) of reusable cups was 13.92%, of which 8.18% were damaged and sent to recycling, and 5.74% were lost (not returned) (see Figure 8). Deposit-refund models (B, C) had lower rates of damaged and lost cups compared to the model when no refund scheme was applied (A) (see Figures 1 and 8). The lowest loss rate was observed in reuse model B1, which was explained by a good combination of economic incentives (fully refundable deposit), governmental support for the reusable solutions in the festival, and good communication strategy. Reuse model C had the second lowest loss rate in comparison with other models, which is explained by the fact that the choice to take a reusable cup was a deliberate action by motivated participants, and a clear message about the reuse model was delivered to the festival participants before the event.

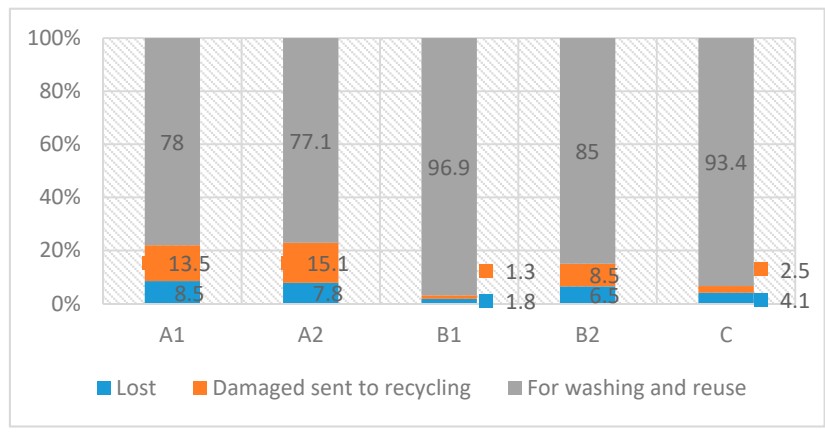

**Figure 8.** Cup damage, loss, and return for reuse rates of different reuse models (A1, A2, B1, B2, C).

Cup loss depends on several factors. An important influence on the results is the number of cups that are taken home [20]. The UK music festival "Shambala" chose to brand 15% of cups, to balance the desire for souvenirs whilst maximizing the reuse to realize environmental benefits [21]. Given the fact that "CupCup" reusable cups had no branding or festival attributes, it did not perform as a souvenir for the participants. Customers' behavior and cooperation level can have a significant impact on the economics of reusable packaging, and are an optimal strategy choice [18]. Reusable packaging damage mainly depends on package design (material used, color, shape, surface texture, etc.), users' behavior, communication strategy, and economic incentives. In this research, the loss of the reusable cups was considered when people took the cups home or threw them out as waste. Most of the cups were lost due to taking them home, and some of the cups lost as waste in the waste bins, especially in the not refundable reuse models (A1, A2). In this research, two

types of damage were distinguished: crucial damage and aesthetic damage (see Figure 9). Damage is crucial when the cups cannot perform their function any longer, such as when they are broken into parts or burnt with cigarettes (see Figure 9a). Aesthetic damage occurs when cups lose visual properties like color, get scratched, bent, or have long-lasting paint or lipstick applied (see Figure 9b). The importance of design, as well as environmental and economic feasibility are presented by Mahmoudi and Parviziomran (2020) as the key measures that have to be addressed to adopt reuse systems. According to Coelho et al. (2020), design can become a limiting factor for reusable packaging system efficiency and sustainability through certain aspects such as system convenience for the customer, material choice, and application of measures for the reduction of product damages and losses.

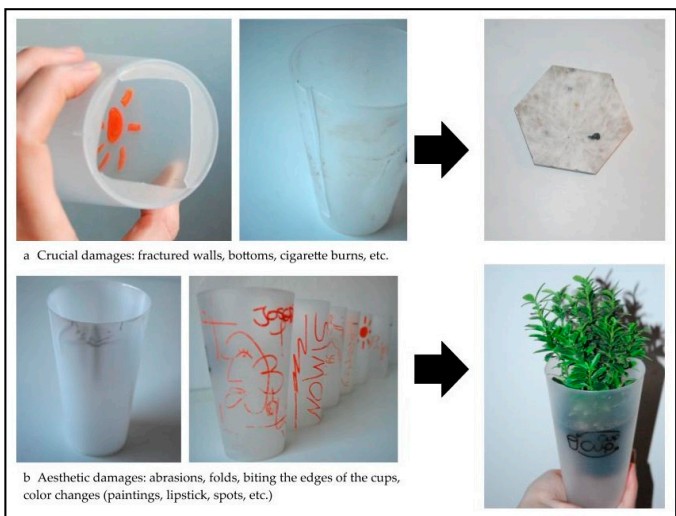

**Figure 9.** Damaged cups from the festivals: (**a**) Cups with crucial damage were recycled; (**b**) Cups with aesthetic damage were repurposed to plant pots.

### 3.1.3. Data Verification

The B2 reuse model was chosen for data verification due to its application for three different events. The verification results show that there is a slight difference in cup loss, damage and return for reuse rates among the three different events where the B2 reuse model was applied, but the differences are not significant (see Figure 10). The data show that total average loss of reusable cups (damaged and lost) for the three festivals was 14.6%, and the average return rate was 85.1%.

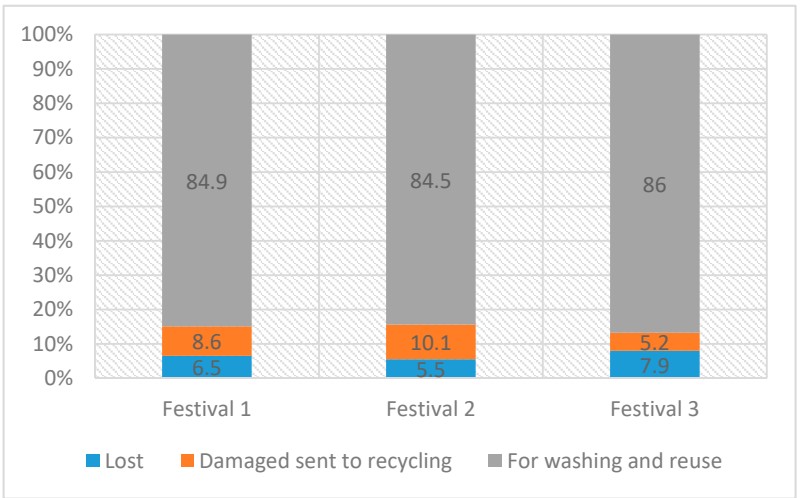

**Figure 10.** Cup damage, loss, and return for reuse rates in three different festivals (1, 2, 3) where the B2 reuse model was applied.

### 3.2. Single-Use Cups Avoided from the Festivals

The amount of eliminated single-use cups from the festivals was measured by multiplying reusable cups used by the average number of drinks consumed by one person per one festival day. The presumption was made that one person consumes 3.5 drinks per one festival day. Figure 11 gives an overview of potential single-use plastic cups avoided from all seven festivals. The results revealed that the total amount of avoided single-use plastic cups from the festivals is 98,280 units. This number corresponds to 786.2 kg of polypropylene waste. Polypropylene is a dominant material for currently used single-use plastic cups for the cold beverages. It was presumed that the collection rate of single-use cups from the festivals was 99%, and used single-use plastic cups are thrown into mixed waste containers and delivered to regional mechanical biological treatment plants (MBTs) together with mixed municipal waste. Lithuania has a regional waste management system with 10 regional waste treatment centers where MBT plants are constructed. All mixed municipal waste is transported to regional MBTs, where sorting is performed in order to separate biological material, solid recovered fuel (fraction for incineration), the fraction suitable for recycling, and the fraction that is only suitable for landfilling.

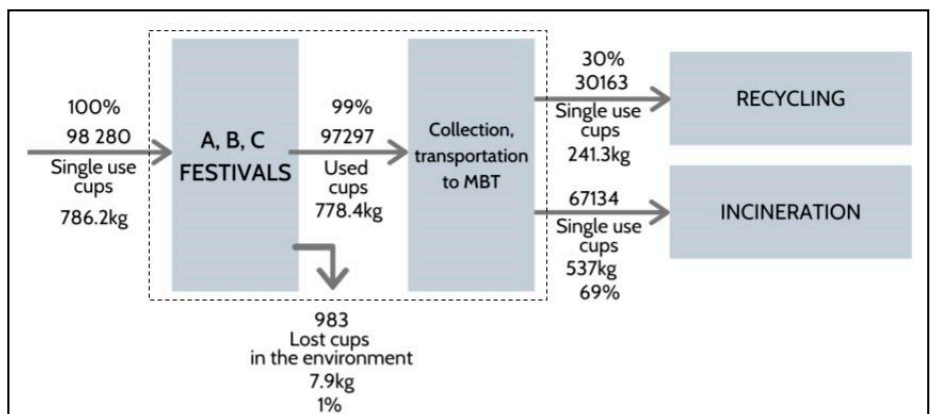

**Figure 11.** Material flow diagram of single-use cups from the festivals, and potential amount of single-use cups avoided from the festivals by applying reusable cup reuse models.

The assumption was made that 69% of single-use plastic cups delivered to MBTs are incinerated together with the combustible waste fraction due to impurities or poor quality. This makes up 537 kg of polypropylene waste, and 30% (241.3 kg) of such plastic waste could be recycled to other products (see Figure 11). The assumption was made based on general statistical information on plastic waste recycling rates that only 30% of plastic packaging is being recycled [9,10]. Packaging waste is collected in two different ways: together with mixed municipal waste (when no sorting is applied); and together with secondary waste (using packaging sorting infrastructure). In the festivals and events, no sorting was performed or only so-called "emotional" sorting was used to demonstrate environmental awareness and responsibility, and therefore, packaging waste such as single-use plastic beverage cups enter mixed municipal waste bins.

Secondary waste that is sorted and collected separately as a part of the EPR (extended producers' responsibility) scheme, is delivered straight to specialized waste sorting centers without entering MBTs. Another assumption was made that 1% of single-use plastic cups are being lost in the environment, and this percentage depends mainly on cleanup quality of the festival area, and participants' behavior.

The findings indicated that 300 cups per 100 participants per one festival day were being avoided in the case of models with the use of reusable cups only (models A, and B); while a mixed reuse model with the usage of both reusable and single-use cups resulted in an avoidance of 108.25 cups per 100 participants per one festival day (model C) (see Figure 12). The numbers show that fully reusable reuse models like models A and B have

higher single-use cup elimination potential than any other mixed systems where both reusable, and single-use options are provided for the customers. Only fully reusable reuse models can achieve material circulation and material savings at a higher rate.

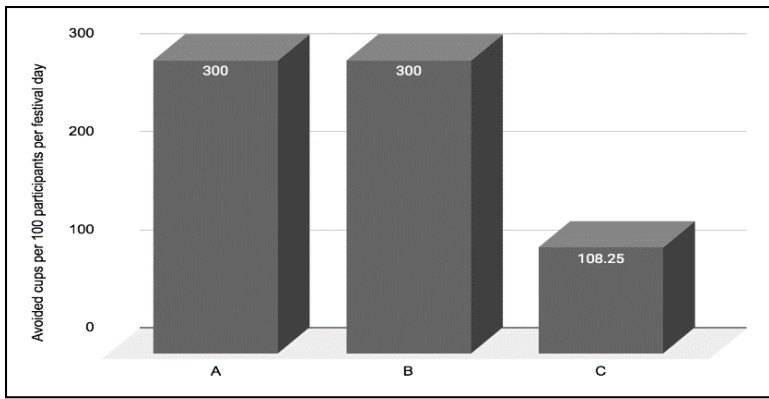

**Figure 12.** Avoided single-use cups per 100 participants per one festival day when different models (A, B, C) applied.

### 3.3. Material Circularity Indicator Results

The results revealed that single-use polypropylene cups with the current waste treatment scenario, where 99% of single-use cups were being collected and treated by recycling or incineration processes, had lower Material Circularity Indicator values compared to reusable cup reuse models. All three reuse models were summarized into one comparable reuse model.

### 3.3.1. Virgin Feedstock

The mass of virgin material (V) was calculated according to Equation (3). The mass (M) of single-use cups is 8 g, while a reusable cup weights 32 g. Both single-use, and reusable cups are produced from 100% virgin source polypropylene, and no recycled content is used for both products ($F_R = 0$). The reusable cup reuse rate is 93% ($F_U = 0.93$), while the single-use cup reuse rate is 0% ($F_U = 0$). The mass of virgin material is therefore: V (reusable cup) = 2.24 g; V (single-use cup) = 8 g.

### 3.3.2. Unrecoverable Waste

The mass of unrecoverable waste (W) was calculated according to Equation (4). It is assumed that 100% of damaged reusable cups are being recycled ($C_R = 1$), and 30% of disposed single-use cups are being recycled ($C_R = 0.3$). The assumption of disposable single-use plastic cup recycling rate (30%) was made according to the currently available statistical data [9,10]. There is no accurate and reliable statistical information on how many single-use plastic packaging is being collected and recycled from festivals and events. A disposable cup recycling rate of 30% was chosen according to some data provided in the Introduction [9,10]. The recycling efficiency rate for polypropylene is assumed to be 99%. The recycling efficiency rate for single-use and reusable cups is $E_C = E_F = 0.99$. The amount of waste generated at the time of collection is: $W_0$ (reusable cup) = 0.32; $W_0$ (single-use cup) = 0.08. The quantity of waste generated in recycling process is: $W_C$ (reusable cup) = 0.2976; $W_C$ (single-use cup) = 0.024. The waste generated to produce the recycled content ($W_F$) is zero. The total amount of unrecoverable waste is: W (reusable cup) = 0.4688; W (single-use cup) = 0.092.

### 3.3.3. Linear Flow Index (LFI)

The LFI was measured according to Equation (2). Linear flow indices for reusable and single-use cups are: LFI (reusable cup) = 0.0423; LFI (single-use cup) = 0.5057. The results

show that the reusable cup linear flow index (LFI) is lower than the single-use cup value, which indicates higher reusability and recyclability of reusable cups.

### 3.3.4. Utility

The utility X was calculated by Equation (5). Average reusable cups were used 105 times (L), $L_{av}$ was around 107 times, U = 1, $U_{av}$ = 1; average single-use cups are used 1 time (L), $L_{av}$ is 1 time, U = 1, $U_{av}$ = 1. The utility X of both reusable and single-use cups are as follows: X (reusable cup) = 0.83; X (single-use cup) = 1.

### 3.3.5. Material Circularity Indicator (MCI)

The MCI was calculated according to Equation (1). The MCI of reusable polypropylene cups was 0.94 and the single-use polypropylene cup MCI was 0.49 (see Figure 13). The calculations show that PP cup reuse has higher MCI than single-use PP cup recycling with 99% collection rate and 30% of recycling rate.

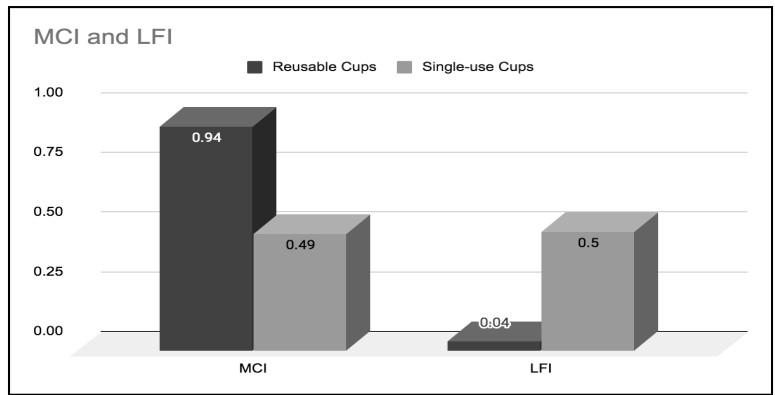

**Figure 13.** Material Circularity Indicator (MCI), and Linear Flow Index (LFI) of reusable polypropylene (PP) cup reuse, and single-use polypropylene (PP) cup recycling.

### *3.4. Overview and Discussion*

Each festival differed in size, time, location, type of drinks served (canned drinks, cocktails, tap drinks), as well as social aspects like average age, education, and dominant social class of attendees. Additionally, all events were affected by some unpredictable circumstances like weather conditions. All of these circumstances can influence the number of cups used, lost, damaged, and returned. Consumers' motivation and awareness are key elements that have a significant impact on the circularity of the packaging.

Waste management at the event or festival is an important factor that determines the overall impact on the environment. It is known that in Lithuania each festival and event is based on linear waste management models. Only a small portion of the events take responsibility and collect waste separately into recycling bins. Sorting is mostly performed due to emotional reasons, and the efficiency of it is questionable. Lithuania has a single-use beverage packaging deposit refund scheme which helps to manage the waste at festivals and reduce the amount of waste. The rest of the waste from take-away food packaging and single-use beverage cups is being thrown away as regular mixed waste. There is a trend in Lithuania to refuse single-use plastic packaging in municipal events and exchange them for the commonly used alternatives—single-use paper-based cups with polyethylene lining as a barrier or bio-based single-use alternatives like paper cups with bioplastic lining or single-use beverage cups from bioplastics, usually from polylactic acid (PLA). These commercially popular biodegradable alternatives require industrial composting equipment which is not available in the Lithuanian market. Thus, this type of single-use paper-based packaging cannot be recycled in Lithuania due to the barrier lining. Most of these single-use plastic alternatives become additional waste and go to MBT plants with the regular mixed waste stream.

Another challenging single-use packaging waste management aspect at the festivals and outdoor events is attendees' behavior, which is hard to influence and control. Sorting skills, knowledge, understanding and awareness level are personal features that have to be developed through societal change over a long period of time. Other factors like package labeling, communication, infrastructure of the recycling stations or sorting bins are also recognized as key aspects to ensure good quality of recovered packaging material and high collection rates.

On the other hand, there is no clear comparative data for how festivals manage their single-use items, and how many of the single-use packages are being recycled. Moreover, a lack of disposable beverage cup waste treatment methods is another issue that needs further study.

Taking into account reusable cup losses, it was noticed that losses depend on several factors and is mainly associated with people taking cups home as souvenirs. Branding of cups can reduce the overall reuse rate and increase environmental impact. Reusable packaging damage mainly depends on package design (material used, color, shape, surface texture, etc.), users' behavior, communication strategy, and economic incentives (Section 3.1.2).

The results are summarized in Table 2.

**Table 2.** Summary overview of the results.

| Festival | 1 | 2 | 3 | 4 | 5 | 6 | 7 |
|---|---|---|---|---|---|---|---|
| Summary of festival type | High standards for attendance, higher price tickets. | Mature attendees, many foreigners. Strict entrance and ticketing policy. | An open-air event at the city center, free of charge, no restricted territory. | Small punk festival. Attendees are younger age, lower entrance fee. | Electronic music festival. Attendees are younger age, higher price tickets, far away from the city. | Electronic and another music festival far away from the city. | Metal, hard rock music. International, many attendees attend it for many years. |
|  | A1 | A2 | B1 | B2 | B2 | B2 | C |
| The applied reuse model for beverage cups | Only reusable cups, non-refundable model: one-time eco-fee with a possibility for unlimited exchange of cups | | Only reusable cups, with deposit-refund | | | | A mixed system of reusable cups with deposit-refund and of single-use cups. |
| Type of refund | No extra fee for a lost cup | Extra fee for a lost cup | Fully refundable | Partly refundable | | | Partly refundable |
| Number of cups per 100 participants per day | 133.33 | 48.66 | 24.51 | 54.02 | | | 36.08 |
| Share of reusable cups, % — For washing and reuse | 78 | 77.2 | 96.9 | 84.9 / 85 | 84.5 / 85 | 86 / 85 | 93.4 |
| Share of reusable cups, % — Damaged, went to recycling | 13.5 | 15.1 | 1.3 | 8.6 / 8.5 | 10.1 / 8.5 | 5.2 / 8.5 | 2.5 |
| Share of reusable cups, % — Lost | 8.5 | 7.8 | 1.8 | 6.5 / 6.5 | 5.5 / 6.5 | 7.9 / 6.5 | 4.1 |

## 4. Conclusions

A study of different models of reusable cup services used during open-air summer festivals to avoid disposable plastic entering the environment demonstrated the difference in achieving results when different models are applied. The three models applied and their two variations reveal the deposit being an important incentive to return more cups while fewer cups are lost or damaged. Still, even in the absence of a deposit, 77.2–78% of the cups were destined for washing and could be reused. The biggest share of cups was returned for reuse under the fully refundable deposit system, almost 97%. Still, such a model has the feature that the costs of providing the service have to be covered by the festival organizers, which is not always possible. With the return of the deposit partially, the problem of financing the service is eliminated and the achieved percentage of reuse of the cups is 85%. A very large proportion of the cups, 93.4%, were returned to reuse when applying the model where festival participants were able to choose between reusable and disposable cups. Still, the overall result of such a model to avoid disposable plastic use is questionable, as only the most conscientious participants chose reusable cups, while others stayed with disposable ones.

There is a lack of studies and information about reusable cup damage and losses during the events, therefore, it is important to compare the study data, and evaluate the efficiency of reusable cup reuse models. The case studies showed that success of the event reuse rate mainly depends on reuse model and communication with festival participants. The research revealed that economic measures have a positive impact on the return rates and preservation of reusable packaging led to higher return rates, and lower damage rates (B, C models). There is a need for a deeper understanding of single-use polypropylene (PP) cup recycling and waste treatment technologies along with methods in the Baltic states region in order to compare single-use and reusable cup systems more precisely. Material and product circularity measurement metrics need development and adaptation to packaging materials and systems. Additionally, it would be useful to include circularity measurements in life cycle assessment metrics. However, in order to assess the full situation and compare reusable models with single-use options, environmental life cycle assessment (LCA) could be used. In addition to LCA, LCC (life cycle costing) may be performed to assess the economic sustainability of reusable cup models vs. single-use cups models.

**Author Contributions:** V.Š. created the research design, performed case studies at seven summer events, collected and analyzed the material, performed calculations, and a literature review. J.K. advised on analysis and presentation of the results, and co-wrote the text of the paper. All authors have read and agreed to the published version of the manuscript.

**Funding:** This research received no external funding. The publication of this article was funded by The Baltic University Programme.

**Institutional Review Board Statement:** Not applicable.

**Informed Consent Statement:** Not applicable.

**Data Availability Statement:** Data available in a publicly accessible repository.

**Conflicts of Interest:** The authors declare no conflict of interest.

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
