# Peer review of "Improvement of Packaging Circularity through the Application of Reusable Beverage Cup Reuse Models at Outdoor Festivals and Events"

_sustainability, doi:10.3390/su13010247_

Round 1

Reviewer 1 Report

The paper describes the performance of service systems implemented in Lithuanian festivals with the aim to identify which models (and market-based instruments) can optimize the use of reusable cups and minimize losses and damages. First, a MFA has been applied to monitor the use and disposal of cups. Then, the MCI has been calculated to monitor the material stream.

The topic seems quite interesting. The paper can provide useful inputs in the field of plastic waste prevention research. However, I detect quite some potential for improvement. The methodology is well described and elaborated but the literature review as well as the implications of the findings are not exhaustive.

Here the list of comments and suggestions for improvement:

The title should be improved. It doesn't capture the main content of the paper.

The introduction is not exhaustive. Please, add data on the consumption of single-use plastic cups and/or waste generated, just to highlight this environmental concern and the need to move from linear to circular economy. Add more information on drivers and challenges characterizing the implementation of reusable packaging models. 

line 50: the scenario you are considering here is not clear. Is the end-of-life? Is the consumption stage or the entire life cycle? please, specify this point

line 50-51: only acronyms have been included. Please,  introduce the full term before introducing the acronyms.

Research questions are not very attractive. They can be considered as sub-research questions. Please, include one more general research question that can underline the importance of implementing reusable models to prevent plastics. 

line 85: you mentioned 29 festivals in the line 71. Please, describe which systems are used in all the festivals or define the 7 festivals as pilot tests or something like that. 

figure 2: landfilling has been considered in figure 1 but not in the calculations. Please, include this End-of-life option or mention the reason why it is not adopted.

Line 381: an assumption has been made on the amount of waste that are sent to the MBT. Could you introduce the criteria used for this assumption? Is it based on real local performance of the waste management system?

Line 377 – 382: when the acronym is introduced, you can used it throughout the paper, thus substituting the full name.

Figure 11: it is not clear the output of this analysis. How have the authors measured the number of avoided cups for each festival? Many elements can affect the results here (number of participants, price of the drinks, type of festivals etc). Have you considered the average number of cups used by one person in a general festival?

Line 416 – 417: many assumptions have been made on the rate of recycling and incineration. Since they strongly affect the results of the MCI, a validation test should be done.

Line 459: In addition to LCA, LCC may be performed to assess the economic sustainability of reusable cups models vs single use cups models.

The outputs of the study basically highlight the performance of the service systems in accordance with the consumer's behaviors and the incentives used. Please, make a critical review on that topic and ends the paper with valuable research outcomes.

Reference: Authors don´t use recent literature. It would be useful to enrich the literature and use references from journals indexed in reputable databases (WoS resp. Scopus).

Corrections should lie in the paper re-formatting according to the instructions included in the review.

Finally, I highly recommend an English native proofreader.

Author Response

Point 1: The title should be improved. It doesn't capture the main content of the paper.

Response 1: Moving Towards more Circular Packaging Systems Through the Implementation of Reusable Beverage Cups Reuse Schemes at Summer Outdoor Festivals and Events.

Point 2: The introduction is not exhaustive. Please, add data on the consumption of single-use plastic cups and/or waste generated, just to highlight this environmental concern and the need to move from linear to circular economy. Add more information on drivers and challenges characterizing the implementation of reusable packaging models. 

Response 2: There is a lack of scientific research made in this area, so the literature is based on available researches and findings. The literature was enriched with scientific data responding to Point 2, as well as reporting to point 14:

Global plastics consumption rates are increasing steadily since the 1950s (Boucher et al., 2019). In 2015 global plastics production has reached 380 million metric tons, with around 40% used for packaging (Gallo et al., 2018). Packaging is recognized as the dominant type of single use plastics in the market, around 60% of all plastic packaging is used for food and beverages market (Groh et al., 2019). The use of plastic packaging is on the rise (Gallo et al., 2018).

Geyer et al. (2017) calculates that in 2015 around 6300 Mt of plastic waste had been generated - 9% recycled, 12% incinerated, and 79% was accumulated in landfills or the natural environment. Authors projects that this linear waste management trend can lead to roughly 12,000 Mt of plastic waste entering the natural environments and landfills by the year of 2050 (Geyer et al., 2017).

Single-use plastics have potential to cause the negative impacts on the environment through littering and accumulation, as well as landfilling, and incineration. Inappropriate disposal of plastics can become a source of leakage to natural environment leading to pollution in many forms (Gallo et al., 2018).

Linear single use plastic consumption model cause wastes leakage into the natural environment, where negative environmental impacts on natural habitats are being caused (Chen et al., 2021).

Due to inadequate management of plastics and plastic waste, some of them end up environment (Boucher et al., 2019).

Every year from 5 to 13 million tonnes of plastic is estimated to end up in the ocean, and plastic pollution reaches even the most remote areas of the planet (Jambeck et al. 2015).

Industry research of UK festivals showed that only 32% of single use PET cups used in UK festivals are recycled (Johnson, C., 2015), and the latest report by European Court of Auditors reveals that new calculations regarding actual share of plastic packaging being recycled should drop down from 42% to 30% across EU member states (European Court of Auditors).

There are certain drivers that can lead to implementation of reusable packaging models, and efficient transition from linear to circular economy, as well as challenges that need to be overcome. The Single-Use Plastics Directive that entered into force on 2 July 2019, is a significant step forward to urge the transition from single-use plastics, towards reusable products and systems (Rethink Plastic alliance and Break Free From Plastic).

The directive foresees certain policy measures, such as market restriction, new product design requirements, extended producer responsibility (EPR) schemes and awareness raising measures, consumption reduction, improved collection and labelling requirements as the key elements to move away from single-use plastic products (Herberz et al. 2020; European Parliament and of the Council, 2019)

Economic incentives can play a big part in reducing the consumption of single-use plastic (Rethink Plastic alliance and Break Free From Plastic).

There are several drivers and factors that can accelerate and improve the transition from a dependence on single-use disposable beverage cups to a sustainable alternative consumption models. There is a strong evidence that charges have a better performance than discounts in reducing the use of single-use disposable beverage cups. A review carried out by the University of Cardif revealed that minimum charge of £0.20 would be needed to change behaviour of 49% of the population. (Poortinga et al. 2020).

Moreover, social marketing tools and information campaigns can contribute to the success of the charges (Poortinga et al. 2020).

The Single-Use Plastics Directive highlights the importance of consumer awareness and information in order to implement the regulatory measures efectivelly, encourage the reduction of onsumption, and redesigning the products.

Increased and improved information and raising consumer awareness go hand-in-hand with the regulatory measures to reduce consumption, redesign products and contribute to ending plastic pollution (Rethink Plastic alliance and Break Free From Plastic).

To reduce environmental impacts, efforts to drastically increase recycling rates of packaging plastics are currently being undertaken (European Parliament).

Boucher, J.; Faure, F.; Pompini, O; Plummer, Z.; Wieser, O; de Alencastro, L. F. (Micro) plastic fluxes and stocks in Lake Geneva basin. TrAC Trends in Analytical Chemistry, 2019, 112, 66-74, doi: 10.1016/j.trac.2018.11.037.

Gallo, F.; Fossi, C.; Weber, R.; Santillo, D.; Sousa, J.; Ingram, I.; Nadal, A.; Romano, D. Marine Litter Plastics and Microplastics and Their Toxic Chemicals Components: the Need for Urgent Preventive Measures. Environmental Sciences Europe 2018, 30, 13, doi.org/10.1186/s12302-018-0139-z.

Jambeck, J.R.; Geyer, R.; Wilcox, C.; Siegler, T.R.; Perryman, M.; Andrady, A.; Narayan, R.; and Law, K.L. Plastic waste inputs from land into the ocean, Science, 2015, 347, 768–771, doi: 10.1126/science.1260352.

Groh, K. J.; Backhaus, T.; Carney-Almroth B.; Geueke B.; Inostroza, P. A.; Lennquist, A.; Leslie, H. A.; Maffini, M.; Slunge, D.; Trasande, L.; Warhurst, A. M.; Muncke. J. Overview of Known Plastic Packaging-Associated Chemicals and Their Hazards. Science of The Total Environment 2019, 651, 3253-3268, doi.org/10.1016/j.scitotenv.2018.10.015.

Geyer, R.; Jambeck, J.R.; Lavender Law, K. Production, Use, and Fate of all Plastics Ever Made. Science Advances 2017, 3, doi:10.1126/sciadv.1700782

Chen, Y.; Awasthi, A.K.; Wei, F.; Tan, O.; Li, J. Single-Use Plastics: Production, Usage, Disposal, and AdverseIimpacts. Science of The Total Environment 2021, 752, doi.org/10.1016/j.scitotenv.2020.141772.

Rethink Plastic alliance and Break Free from Plastic. Moving away from single-use. Guide for national decision makers to implement the single-use plastics directive. Report, 2019. Available online: https://rethinkplasticalliance.eu/wp-content/uploads/2019/10/2019_10_10_rpa_bffp_sup_guide.pdf (accessed on 2020-11-19).

Herberz, T.; Barlow, C.Y.; Finkbeiner, M. Sustainability Assessment of a Single-Use Plastics Ban. Sustainability 2020, 12, 3746, https://doi.org/10.3390/su12093746.)

European Parliament and of the Council. Directive 2019/904 on the reduction of the impact of certain plastic products on the environment. Official Journal of the European Union 2019. Available online: https://eur-lex.europa.eu/legal-content/EN/TXT/PDF/?uri=CELEX:32019L0904)

Poortinga, W.; Nash, N.; Hoeijmakers, L. Rapid Review of Charging for Disposable Coffee Cups and other Waste Minimisation Measures. 2019. Available online: http://orca.cf.ac.uk/124422/1/rapid-review-charging-disposable-coffee-cups-waste-minimisation-measure-full-report.pdf (accessed on 2020-11-19).

Johnson, C. The Show Must Go On. Environmental Impact Report and Vision for the UK Festival Industry. Powerful Thinking. 2015. Available online: http://www.powerful-thinking.org.uk/site/wp-content/uploads/The-Show-Must-Go-On-Report.pdf (accessed on 2020-11-21).

European Court of Auditors. Plastic packaging waste: EU needs to boost recycling to achieve ambitions. Press Release, Luxembourg, 6 October 2020. Available online: https://www.eca.europa.eu/Lists/ECADocuments/INRW20_04/INRW_Plastic_waste_EN.pdf (accessed on 2020-11-21).

Point 3: line 50: the scenario you are considering here is not clear. Is the end-of-life? Is the consumption stage or the entire life cycle? please, specify this point.

Response 3: The text was improved this way:

From: “Regarding reusable cup models, they display a lower environmental burden than those of disposable cups despite the fact that life cycle assessment (LCA) for reusable cups were always based on the most disadvantageous scenario [5].”.

To: “A comprehensive comparative life cycle analysis of different cup systems for selling of drinks at events, where different cup systems and materials were examined. The study revealed that reusable cup models have lower environmental burden comparing to disposable cup models, despite the fact that life cycle assessment (LCA) for reusable cups were always based on the most disadvantageous scenario comparing to disposable cups [5].”.

Point 4: line 50-51: only acronyms have been included. Please, introduce the full term before introducing the acronyms.

Response 4: The text was improved this way: life cycle assessment (LCA); Realising Another World (RAW) Foundation; polyethylene terephthalate (PET).

Point 5: Research questions are not very attractive. They can be considered as sub-research questions. Please, include one more general research question that can underline the importance of implementing reusable models to prevent plastics. 

Response 5: The aim of the research was specified, and linked with the research questions:

“The objective of this work is to assess the impact of the application of cup reuse models in open summer festivals on material circularity and the avoidance of disposable plastic.

The paper describes the performance of service systems implemented in Lithuanian festivals with the aim to identify which models (and market-based instruments) can optimize the use of reusable cups and minimize losses and damages.

The aim of this work was achieved by answering three research questions (RQs):

  1. How many cups will be used, lost, and damaged during the festivals when different reusable cups reuse models are applied (RQ1)?
  2. What impact do reusable cup reuse models make on material circularity compared to the use of single-use beverage cups at the festivals (RQ2)?
  3. How many single-use plastic cups were avoided by applying reusable beverage cup reuse models at the festivals (RQ3)?

First, a material flow analysis (MFA) has been applied to monitor the use and disposal of cups. Then, the material circularity indicator (MCI) has been calculated to monitor the material stream.”

Point 6: line 85: you mentioned 29 festivals in the line 71. Please, describe which systems are used in all the festivals or define the 7 festivals as pilot tests or something like that. 

Response 6: certain changes in line 71, and line 85 have been made:

Line 71:

From: “29 festivals and festival-like events were organized during the 2019 summer season in Lithuania.”

To: “29 festivals and festival-like events were organized during the 2019 summer season in Lithuania where standard linear “take-make-dispose” model of disposable single-use beverage cups was applied. No clear measures have been made before in order to minimise the use of disposable beverage cups, and reduce the waste of single-use plastics within the events. Until then, event organizers had no alternative to disposable beverage. Several events used reusable cup deposit systems in the past, but the cups for these events had promotional attributes and were therefore used as souvenirs.”.

Line 85:

From: “Three alternative reusable beverage cup reuse models were tested at 7 Lithuanian open-air festivals during the summer season of 2019 in order to eliminate single-use plastic cups.”.

To: “Three alternative reusable beverage cup reuse models were implemented as pilot tests at 7 Lithuanian open-air festivals during the summer season of 2019 in order to eliminate single-use plastic cups.”.

Point 7: figure 2: landfilling has been considered in figure 1 but not in the calculations. Please, include this End-of-life option or mention the reason why it is not adopted.

Response 7: Thanks for the notice, the landfilling was really missing, and changes have been made.

Figure 2. A simplified scheme of reusable and single-use cups flows.

Point 8: Line 381: an assumption has been made on the amount of waste that are sent to the MBT. Could you introduce the criteria used for this assumption? Is it based on real local performance of the waste management system?

Response 8: The assumption is made based on general statistical information on the plastic waste recycling rates, which is known that only 30% of plastic packaging is being recycled. Packaging waste is collected in two different ways: together with mixed municipal waste (when no sorting is applied); and together with secondary waste (using packaging sorting infrastructure). In the festivals and events no sorting is performed or only so called “emotional” sorting is used to demonstrate environmental awareness and responsibility, so packaging waste such as single-use plastic beverage cups enter mixed municipal waste bins.

Lithuania has regional waste management system with 10 regional waste treatment centers, where MBT plants are constructed. All mixed municipal waste is transported to regional MBTs, where sorting is performed in order to separate biological fraction, solid recovered fuel (fraction for incineration), fraction suitable for recycling, and fraction that is only suitable for landfilling.

Secondary waste that is sorted and collected separately as a part of EPR (extended producers’ responsibility) scheme, is delivered straight to specialized waste sorting centers without entering MBTs.  

Point 9: Line 377 – 382: when the acronym is introduced, you can used it throughout the paper, thus substituting the full name.

Response 9: “Mechanical biological treatment plant” was substituted as a full name, only acronym MBT left.

Point 10: Figure 11: it is not clear the output of this analysis. How have the authors measured the number of avoided cups for each festival? Many elements can affect the results here (number of participants, price of the drinks, type of festivals etc). Have you considered the average number of cups used by one person in a general festival?

Response 10: Line 369-line371: “The amount of eliminated single-use cups from the festivals was measured by multiplying reusable cups used by the average number of drinks consumed by one person per one festival day. The presumption was made that one person consumes 3.5 drinks per one festival day).”

Point 11: Line 416 – 417: many assumptions have been made on the rate of recycling and incineration. Since they strongly affect the results of the MCI, a validation test should be done.

Response 11: The assumption was made according to the current available statistical data. There is no accurate and reliable statistical information of how many single-use plastic packaging is being collected and recycled from the festivals, and events – no studies have been made in order to calculate the real input of single-use packaging to the event, and the real output of collected and recycled single-use packaging from the events. The same situation exists on the market – there is a gap in the packaging waste accounting, collection and recycling rate accuracy. Additional investigations need to be made in order to gather the real data of single-use packaging collection and recycling rates, especially from certain events and festivals.

Due to this systematic problem, only assumptions were used in order to compare the circularity metrics among single-use, and reusable packaging. Disposable cups recycling rate (30%) was chosen according to some data:

Industry research of UK festivals showed that only 32% of single use PET cups used in UK festivals are recycled (Johnson, C., 2015), and the latest report by European Court of Auditors reveals that new calculations regarding actual share of plastic packaging being recycled should drop down from 42% to 30% across EU member states (European Court of Auditors).

In section “Conclusions” it was mentioned that the article has some drawbacks regarding accuracy of data, and the need of factual plastic packaging recycling rate, and situation: “There is a need for a deeper understanding of single use polypropylene (PP) cups recycling and waste treatment technologies and methods in Baltic states region in order to compare single-use and reusable cups system more precisely.”.

Johnson, C. The Show Must Go On. Environmental Impact Report and Vision for the UK Festival Industry. Powerful Thinking. 2015. Available online: http://www.powerful-thinking.org.uk/site/wp-content/uploads/The-Show-Must-Go-On-Report.pdf (accessed on 2020-11-21).

European Court of Auditors. Plastic packaging waste: EU needs to boost recycling to achieve ambitions. Press Release, Luxembourg, 6 October 2020. Available online: https://www.eca.europa.eu/Lists/ECADocuments/INRW20_04/INRW_Plastic_waste_EN.pdf (accessed on 2020-11-21).

Point 12: Line 459: In addition to LCA, LCC may be performed to assess the economic sustainability of reusable cups models vs single use cups models.

Response 12:

From: “However, in order to assess the full situation, and compare reusable models with single use options, environmental life cycle assessment and economic feasibility studies would also provide additional value.”

To: “However, in order to assess the full situation, and compare reusable models with single-use options, environmental life-cycle assessment (LCA) could be used.  In addition to LCA, LCC (life-cycle costing) may be performed to assess the economic sustainability of reusable cups models vs single use cups models.”.

Point 13: The outputs of the study basically highlight the performance of the service systems in accordance with the consumer's behaviors and the incentives used. Please, make a critical review on that topic and ends the paper with valuable research outcomes.

Response 13: Conclusions paragraph was improved by critical review on the topic, and results.

Point 14: Reference: Authors don´t use recent literature. It would be useful to enrich the literature and use references from journals indexed in reputable databases (WoS resp. Scopus).

Response 14: The literature was enriched with literature responding to Point 2.

Reviewer 2 Report

Dear Authors and Editors

The manuscript considering the waste management in the summer festival. It is interesting topic and excellent way contribute to the parcticalities of circular economy, and make it more acceptable within concumers. 

However, I see some lackings in the manuscript, which could be improved improve before publications.

The aim of this work should be rewritten. It could be more clearer.

More information about the festival could be explained, e.g. style (classic music festival or rock concert) and what kind audience were there? It has a significance on the behavior with waste.

line 185: What means LFI? ...Abbreviation could be connected with the terminology (linear flow index) at first time whn it is menitoned

Implications of the work should be described more clearly.

I'm afraid that reference list wasn't done according to the style of journal. Please check it carefully.

...I'm wondering, if the work have done during summer 2019, why you will publish just now, why a year ago?

Numbers within keywords are useless. Please remove it.

Author Response

Point 1: The aim of this work should be rewritten. It could be more clearer.

Response 1: The aim of the research was specified, and linked with the research questions:

“The objective of this work is to assess the impact of the application of cup reuse models in open summer festivals on material circularity and the avoidance of disposable plastic.

The paper describes the performance of service systems implemented in Lithuanian festivals with the aim to identify which models (and market-based instruments) can optimize the use of reusable cups and minimize losses and damages.

The aim of this work was achieved by answering three research questions (RQs):

  1. How many cups will be used, lost, and damaged during the festivals when different reusable cups reuse models are applied (RQ1)?
  2.  What impact do reusable cup reuse models make on material circularity compared to the use of single-use beverage cups at the festivals (RQ2)?
  3. How many single-use plastic cups were avoided by applying reusable beverage cup reuse models at the festivals (RQ3)?

First, a material flow analysis (MFA) has been applied to monitor the use and disposal of cups. Then, the material circularity indicator (MCI) has been calculated to monitor the material stream.”

Point 2: More information about the festival could be explained, e.g. style (classic music festival or rock concert) and what kind audience were there? It has a significance on the behavior with waste.

Response 2: Additional information about each festival is provided in Table 1, where certain aspects like time and location, average age of attendees, and main concept of the festival was provided.

Table 1. Summary of the festivals, the applied reuse models, and additional characteristics of each festival.

Festival

1

2

3

4

5

6

7

Number of attendees

1500

900

3000

500

950

700

2000

Duration, days

3

3

4

2

3

2

3

The applied reuse model for beverage cups

A1

A2

B1

B2

B2

B2

C

Festival characteristics (time, location)

July 4-7, Palanga

August 8-11, Trakai district

28 - September 1, Nida

June 14-16, Vilnius region

July 25-28, Utena region

August 9-11, Trakai district

July 11-14, Anykščiai

Average age of attendees (years)

30+

30+

35+

20+

20+

25+

25+

Main concept, activities

Special guests, high standards for attendance: only with costumes, higher price tickets, secret and very secured festival – party.

Mature attendees; over 50% foreigners; most of them artists, hipsters, creative people. The festival had strict entrance and ticketing policy- organizers choose to whom sell the tickets.

Openair event at the city centre, free of charge, no restricted territory or ticketing. Live music, food, drinks, yacht racing.

Small punk festival, alternative, experimental music. The main concept of the festival was green and sustainable event. Attendees are younger age, lower entrance fee.

Electronic music festival. Strict face control, people are long-term participants, the higher price tickets, far away from city, in the middle of woods.

Electronic music festival, dark music, post punk, lot of DJ sets, some experimental music, people are long-term participants, far away from city, surrounded by nature.

Metal, hard rock music, niche festival, international, popular among Baltic states hard rock fans. Has its own audience, most attendees are fans of this festival and attend it many years as a tradition. 

Point 3: Line 185: What means LFI? ...Abbreviation could be connected with the terminology (linear flow index) at first time whn it is menitoned

Response 3: LFI as linear flow index is explained in line 182 (below the equation).

Point 4: Implications of the work should be described more clearly.

Response 4: The results and outcomes of the research are discussed in Conclusions section more clearly.

Point 5: I'm afraid that reference list wasn't done according to the style of journal. Please check it carefully.

Response 5: The style of the reference list was improved according to the journal requirements as well as enriched with some additional scientific literature.  

References

  1. Zelenika, I.; Moreau, T.; Zhao, J. Toward Zero Waste Events: Reducing Contamination in Waste Streams with Volunteer Assistance. Waste Management, 2018, 76, 39-45, doi: 10.1016/j.wasman.2018.03.030.
  2. Martinho, G.; Gomes, A.; Ramos, M.; Santos, P.; Gonçalves, G.; Fonseca, M.; Pires, A. Solid Waste Prevention and Management at Green Festivals: A Case Study of the Andanças Festival, Portugal. Waste Management, 2018, 71, 10-18, doi: 10.1016/j.wasman.2017.10.020.
  3. 3. Van der Wagen, L. Events & Tourism Essentials. 1st; Pearson Australia, 2010, pp. 65-80, ISBN 9781442517837.
  4. Rigamonti, L.; Biganzoli, L.; Grosso, M. Packaging Re-use: a Starting Point for its Quantification. Journal of Material Cycles and Waste Management 2019, 21, 35-43, doi: 10.1007/s10163-018-0747-0.
  5. Pladerer, C.; Meissner, M.; Dinkel, F.; Zschokke, M.; Dehoust, G.; Schüler, D. Comparative Life Cycle Assessment of Various Cup Systems for the Selling of Drinks at Events. Available online: http://www.meucopoeco.com.br/environmental_study.pdf (accessed on 2 November 2020).
  6. Realising Another World (RAW) Foundation. The Drastic on Plastic Reusable Bar Cup Guide. Available online: http://www.powerful-thinking.org.uk/site/wp-content/uploads/RAW_REUSABLE-CUPS-GUIDE.pdf  (accessed on 2 November 2020).
  7. Garrido N.; Castillo, A. Environmental Evaluation of Single-Use and Reusable Cups. The International Journal of Life Cycle Assessment 2007, 12, 252-256.
  8. European Commission. Staff Working Document. Impact Assessment. Reducing Marine Litter: Action on Single-use Plastics and Fishing Gear. Brussels, 2018. Available online: https://ec.europa.eu/environment/circular-economy/pdf/single-use_plastics_impact_assessment.pdf (accessed on 2 November 2020).
  9. Linder, M.; Sarasini, S.; Loon. P. A Metric for Quantifying Product-Level Circularity. Journal of Industrial Ecology 2017, 21, 545–558, doi: 10.1111/jiec.12552.
  10. 10. European Commission. Communication from the Commission to the European Parliament, the Council, the European Economic and Social Committee and the Committee of the Regions. A New Circular Economy Action Plan. For a Cleaner and More Competitive Europe. Brussels, 2020.
  11. 11. Zainal, Z. Case study as a Research Method. Jurnal Kemanusiaan 2007, 9. Available online: https://core.ac.uk/download/pdf/11784113.pdf (accessed on 5 October 2020).
  12. 12. Cencic, O; Rechberger, H. Material Flow Analysis with Software Stan. Environmental Engineering and Management Journal 2008, 18, 3-7.
  13. 13. Moriguchi, Y. Material Flow Indicators to Measure Progress Toward a Sound Material-Cycle Society. Journal of Material Cycles and Waste Management 2007, 9, 112–120, doi:10.1007/s10163-007-0182-0.
  14. Ellen MacArthur Foundation. Circularity Indicators. An Approach to Measuring Circularity. Project Overview. Available online: https://www.ellenmacarthurfoundation.org/assets/downloads/Circularity-Indicators-Methodology.pdf (accessed on 5 October 2020).

Point 6: I'm wondering, if the work have done during summer 2019, why you will publish just now, why a year ago?

Response 6: It took quite a time to finalize the research after intensive 2019 summer season, make accounting for damaged cups, and decide how to characterize damages, calculate the data, prepare the poster for the BUP Symposium, and then choose the journal for publication. The data after this summer season is not prepared yet.

Point 7: Numbers within keywords are useless. Please remove it.

Response 7: The numbers of the keywords have been removed.  

Reviewer 3 Report

In common: interesting research that  needs to be published. But the papers asks some revision.

In the conclusions nothing from the found figures of the research is taken up, only rather common remarks about topics that need further research. At first I would expect an overview of found results of researched festivals. Together with what this means for the mass balance. Many of the mentioned topics in the conclusion can better be taken up in a paragraph discussion like lack of insight in waste treatment, how festivals treat single use cups etc.

A lot of influencing parameters for the return rate are mentioned like type of event, average number age of visitors to the festival, influence of the drink providing companies, branding and/or printing the cups. The same accounts for damaged cups. These parameters appear in the text at points where they are of relevance. It would be better to present an overview of all these influencing factors and appoint to that overview (and/or take it up in the overview of type of festivals). At line 350 the paragraph show these influences.

Some details:

Summary: The text of scenario B does not look right

Line 225 end with “:”, there should be an equation.

Line 267. The units in figure 3 in the black areas are not presented. Better to mention them one time.

Line 309: From 6000 participants only 309 2000 participants chose reusable “CupCup” alternative. 1/3 is more than ‘only’.

Line 315. Figure 7, figures have one figure behind the comma while in other figures these were 2.

Line 373. 98 280. This means 98 thousand, 2 hundred and eighty?

Author Response

Point 1: In the conclusions nothing from the found figures of the research is taken up, only rather common remarks about topics that need further research. At first I would expect an overview of found results of researched festivals. Together with what this means for the mass balance. Many of the mentioned topics in the conclusion can better be taken up in a paragraph discussion like lack of insight in waste treatment, how festivals treat single use cups etc.

Response 1: The results of the festival together with mass balance interpretation was delivered in Conclusions paragraph, while some of the topics discussed in Conclusions section was transferred to Results and discussion paragraph in a new section with additional information was provided:

Each festival and event are based on linear waste management model. Only a small part of the events take responsibility and collect waste separately into recycling bins. Sorting is mostly performed due to emotional reasons, and the efficiency of it is questionable. Lithuania has single-use beverage packaging deposit refund scheme which helps to manage the waste at the festivals, and reduces waste amounts. The rest of the waste from take-away food packaging and single-use beverage cups are being thrown away as a regular mixed waste. There is a trend in Lithuania to refuse single-use plastic packaging in municipality events, and exchange them to the commonly used alternatives - single-use paper-based cups with polyethylene lining or as a barrier or bio based single-use alternatives, like paper cups with bioplastic lining or single-use beverage cups from bioplastics, usually from polylactic acid (PLA). These commercially popular biodegradable alternatives require industrial composting equipment which is not available in Lithuanian market. Thus, this type of single-use paper-based packaging cannot be recycled in Lithuania as well due to barrier lining. Most of these single-use plastic alternatives become additional waste and go to MBT plants with a regular mixed waste stream.

Another challenging single-use packaging waste management aspect at the festivals and outdoor events is attendees’ behavior, which is hard to influence and control. Sorting skills, knowledge, understanding and awareness level are personal features that has to be developed in a longer perspective together with societal changes. Other factors like packaging labeling, communication, infrastructure of the recycling stations or sorting bins are also recognized as key aspects to ensure good quality of recovered packaging material and high collection rates.

Point 2: A lot of influencing parameters for the return rate are mentioned like type of event, average number age of visitors to the festival, influence of the drink providing companies, branding and/or printing the cups. The same accounts for damaged cups. These parameters appear in the text at points where they are of relevance. It would be better to present an overview of all these influencing factors and appoint to that overview (and/or take it up in the overview of type of festivals). At line 350 the paragraph show these influences.

Response 2: All suggested influencing factors and other relevant data from Conclusions paragraph were transferred to section no. 3.4.

Point 3: Summary: The text of scenario B does not look right

Response 3: The text was improved this way:

From: “This model provided partly or fully refundable reuse schemes for attendees:

B1. Fully refundable type of event refunded full deposit price for the attendees. This model is easy to communicate, nevertheless the costs of the service has to be funded by organizers;

B2. Partly refundable type of event refunded half of the price attendees paid, in order to cover service costs from the fee of attendees. B2 model was used for data verification. The following reuse model was applied in 3 events, so this particular model was used for data verification in order to check data reliability. Verification is used in order to identify how significant cups loss, damage, and return rates differ within 3 different events with same reuse model (B2).”.

To: “This model provided partly, and fully refundable reuse schemes for events attendees:

B1. Fully refundable reuse model. Full deposit was refunded for the attendees. This model is easy to apply, communicate, nevertheless the costs of the service has to be covered by organizers, in contrast with B2 reuse model, where half of the amount payed by attendees is used as a service fee to cover costs, and another half is used as a deposit for better cups returns;

B2. Partly refundable reuse mode. Only half of the amount was refunded when the cup was returned. The non-refundable part of the deposit was considered as a fee in order to cover service costs, and the other half was refundable deposit that motivated attendees to return the cups. B2 model was used for data verification. 3 events applied the following reuse model, so this particular B2 model was chosen to verify the data, and check data reliability. Verification is used in order to identify how significant cups loss, damage, and return rates differ within 3 different events with the same reuse model (B2) applied.”.

Point 4: Line 225 end with “:”, there should be an equation.

Response 4: “The utility X is expressed by the following equation:” was removed because the equation was used above, and the sentence was left as a mistake.

Point 5: Line 267. The units in figure 3 in the black areas are not presented. Better to mention them one time.

Response 5: The figure in the black area is expressed as number of cups for 100 participants per 1 festival day. The corrections were made in all material flow diagrams (figure 3, figure 4, figure 5, figure 6, figure 7), and the flows were explained by this expression:

Point 6: Line 309: From 6000 participants only 309 2000 participants chose reusable “CupCup” alternative. 1/3 is more than ‘only’.

Response 6: The text was improved this way:

From: “From 6000 participants only 2000 participants chose reusable “CupCup” alternative.”.

To: “From 6000 participants 2000 participants chose reusable “CupCup” alternative, which accounts for 1/3 of total festival attendees.”.

Point 7: Line 315. Figure 7, figures have one figure behind the comma while in other figures these were 2.

Response 7: In Figure 7, three figures (34.6; 0.9; 33.7) have one figure behind the comma due to the first number behind the comma is round (6; 9; 7):

100 people/

1 day

%

36.08

100.00

1.48

4.11

34.60

95.89

0.90

2.49

33.70

93.39

Point 8: Line 373. 98 280. This means 98 thousand, 2 hundred and eighty?

Response 8: Yes, the number is 98 thousand, 2 hundred and eighty.

Round 2

Reviewer 1 Report

The paper has been significantly improved. It may massively contribute to increasing the quality of the research on reusable beverage cups system.

Please, consider the last suggestions listed below before going to the publication:

  • The title seems too long. It is not mandatory but it may be summarized. 
  • Please, keep attention to the decimal separator (line 42 and line 44)
  • The content of the sentence reported in line 61 is not contextualized. You may first introduce the fact that in some cases waste collection is missing or difficult to implement and consequently, waste recycling performances are low. This is the case of festivals etc. (line 62)
  •  The Commission has already reviewed the Packaging and packaging waste directive. See Directive (EU) 2018/852 (line 64)
  • Introduce the new Circular economy action plan to highlight the intention of the Commission to move forwards reuse over recycling strategies. See Eleonora Foschi, Sara Zanni, Alessandra Bonoli. (2020). Combining Eco-Design and LCA as Decision-Making Process to Prevent Plastics in Packaging Application. In Sustainability journal.
  • Introduce a brief description of the state of the art of reuse models in packaging applications and their added value in specific context. See a. Mahmoudi, M., & Parviziomran, I. (2020). Reusable packaging in supply chains: A review of environmental and economic impacts, logistics system designs, and operations management. In International Journal of Production Economics; b. Coelho, P. M., Corona, B., ten Klooster, R., & Worrell, E. (2020). Sustainability of reusable packaging–Current situation and trends. In Resources, Conservation and Recycling journal; c. Cottafava, D., Costamagna, M., Baricco, M., Corazza, L., Miceli, D., & Riccardo, L. E. (2020). Assessment of the environmental break-even point for deposit return systems through an LCA analysis of single-use and reusable cups. Sustainable Production and Consumption journal.
  • Move from "to assess the impact of" to "to assess the environmental benefits of"
  • Include references and sources (line 437 and lines 482-485)
  • Review the language

Author Response

Point 1: The title seems too long. It is not mandatory but it may be summarized.

Response 1: The title has been changed this way:

From: “Moving Towards More Circular Packaging Systems Through Implementation of Reusable Beverage Cups Reuse Schemes at Outdoor Summer Festivals and Events”.

To: “Improvement of Packaging Circularity through the Application of Reusable Beverage Cup Reuse Models at Outdoor Festivals and Events”.

Point 2: Please, keep attention to the decimal separator (line 42 and line 44).

Response 2: The comma was left as it is, no changes have been made.

Point 3: The content of the sentence reported in line 61 is not contextualized. You may first introduce the fact that in some cases waste collection is missing or difficult to implement and consequently, waste recycling performances are low. This is the case of festivals etc. (line 62).

Response 3: The changes have been made:

From: The latest report by the European Court of Auditors reveals that new calculations regarding the actual share of plastic packaging being recycled should drop down from 42% to 30% across EU member states [9]. Industry research of UK festivals showed that only 32% of single-use PET cups used in UK festivals are recycled [10].

To: The latest report by the European Court of Auditors reveals that new calculations regarding the actual share of plastic packaging being recycled should drop down from 42% to 30% across EU member states [9]. In some cases, waste collection is missing or is difficult to implement, and consequently, waste recycling performances are low. This is the case of the festivals. Industry research of UK festivals showed that only 32% of single-use PET cups used in UK festivals are being recycled [10].

Point 4: The Commission has already reviewed the Packaging and packaging waste directive. See Directive (EU) 2018/852 (line 64).

Response 4: The changes have been made:

From: European Commission is about to review Directive 94/62/EC to reinforce the mandatory essential <…>

To: European Commission has reviewed Directive 94/62/EC to reinforce the mandatory essential <…>

Point 5: Introduce the new Circular economy action plan to highlight the intention of the Commission to move forwards reuse over recycling strategies. See Eleonora Foschi, Sara Zanni, Alessandra Bonoli. (2020). Combining Eco-Design and LCA as Decision-Making Process to Prevent Plastics in Packaging Application. In Sustainability journal.

Response 5: Changes have been made among lines 64-72, and some highlights about Circular economy action plan had been added using Eleonora Foschi, Sara Zanni, Alessandra Bonoli. (2020) article:

From: European Commission is about to review Directive 94/62/EC to reinforce the mandatory essential requirements for packaging to be allowed on the EU market with a focus on reduction of packaging and packaging waste through waste prevention measures, and design for re-use and recyclability of packaging [11]. The Single-Use Plastics Directive that entered into force on 2 July 2019 is another significant step forward to urge the transition from single-use plastics, towards reusable products and systems [12]. The directive foresees certain policy measures, such as market restriction, new product design requirements, extended producer responsibility (EPR) schemes, and awareness-raising measures, consumption reduction, improved collection and labeling requirements as the key elements to move away from single-use plastic products [8, 13].

To: European Commission has reviewed Directive 94/62/EC to reinforce the mandatory essential requirements for packaging to be allowed on the EU market with a focus on reduction of packaging and packaging waste through waste prevention measures, and design for re-use and recyclability of packaging [11]. A new Circular Economy Action Plan, represented by the European Commission in 2020, also aims to accelerate the transition from materials recycling to waste prevention and reuse strategies (Eleonora et al. (2020); [17]). The Plan gives incentives that pursue the transformation of current packaging systems to more circular and sustainable ones (Eleonora et al. (2020); European Commission (2020).

The Single-Use Plastics Directive that entered into force on 2 July 2019 is another significant step forward to urge the transition from single-use plastics, towards reusable products and systems [12]. The directive foresees certain policy measures, such as market restriction, new product design requirements, extended producer responsibility (EPR) schemes, and awareness-raising measures, consumption reduction, improved collection and labeling requirements as the key elements to move away from single-use plastic products [8, 13].

Point 6: Introduce a brief description of the state of the art of reuse models in packaging applications and their added value in specific context. See

  1. Mahmoudi, M., & Parviziomran, I. (2020). Reusable packaging in supply chains: A review of environmental and economic impacts, logistics system designs, and operations management. In International Journal of Production Economics;
  2. Coelho, P. M., Corona, B., ten Klooster, R., & Worrell, E. (2020). Sustainability of reusable packaging–Current situation and trends. In Resources, Conservation and Recycling journal;
  3. Cottafava, D., Costamagna, M., Baricco, M., Corazza, L., Miceli, D., & Riccardo, L. E. (2020). Assessment of the environmental break-even point for deposit return systems through an LCA analysis of single-use and reusable cups. Sustainable Production and Consumption journal.

Response 6: Additional description about reuse models in packaging has been added:

Relatively limited use and application of reusable packaging reuse systems today might be the result of the global drift from reusable packaging to single-use packaging during the past decades (Coelho et al. (2020).

Nevertheless, Coelho et al. (2020) claim that packaging reuse is not new in both B2B (business to business), and B2C (business to consumer) segments, where the application of different forms of reusable packaging solutions have been established from primary, secondary, to tertiary (transportation) packaging (Mahmoudi & Parviziomran (2020).

Mahmoudi & Parviziomran (2020) points out that even reuse practices are not new, but primary packaging, that has direct contact with the product, is recognized to be a newer concept in comparison to secondary and tertiary packaging reusability.

Also, changes in packaging ownership have been observed - the author claims that there is a clear move from packaging owning to renting services (Mahmoudi & Parviziomran (2020)). This could have a significant impact on our consumption patterns.

Coelho et al. (2020) also assert the growth of acceptance for service systems, instead of ownership in the reusable packaging market. This ownership transformation has both some evidential advantages and challenges for the decision-makers.

As the authors have noted, reusable packaging return systems may increase customers' loyalty to retailers through refund schemes. Authors see advantages for the reusable returnable packaging consumers, which are cost reduction through discounts, as well as the reduction in waste amounts (Coelho et al. (2020).

The majority of authors agree that reuse of packaging has a huge potential for environmental impact reduction through the decreased need of materials, especially virgin or primary materials when the reuse systems are well developed and function sustainably (Coelho et al. (2020); Cottafava et al. (2020)).

Nevertheless, reuse packaging systems may have a higher environmental footprint compared to single-use packaging if the reuse systems are not being managed carefully in a sustainable way. Most of the negative impacts of reusable packaging reuse systems have the potential to occur through certain aspects such as inappropriate packaging design, complex logistics, containers’ service life, cleaning, food safety, etc. ((Mahmoudi & Parviziomran (2020); Coelho et al. (2020)).

Coelo et al. (2020) agree that economics, as well as environmental impacts of reuse systems strongly, depend on several factors like logistics, reusable packaging return rates, cleaning, labor costs, and market size. The author notes that more studies of life cycle costing have to be made to compare the costs of single-use and reusable packaging systems.

Some supplementary literature was added to Results section:

Customers’ behavior and cooperation level can have a significant impact on the economics of reusable packaging, and optimal strategy choice (Mahmoudi, M., & Parviziomran, I. (2020).

The importance of design, as well as environmental and economic feasibility are presented by Mahmoudi, M., & Parviziomran, I. (2020) as the key measures that have to be addressed to adopt reuse systems.

According to Coelho et al. (2020), design can become a limiting factor for reusable packaging system efficiency and sustainability through certain aspects such as system convenience for the customer, material choice, application of measures for the reduction of product damages and losses.

Point 7: Move from "to assess the impact of" to "to assess the environmental benefits of".

Response 7: The changes have been made:

From: The objective of our work was to assess the impact of the application of cup reuse models in open summer festivals on material circularity and the avoidance of disposable plastic.

To: The objective of our work was to assess the environmental benefits of the application of cup reuse models in open summer festivals on material circularity and the avoidance of disposable plastic.

Point 8: Include references and sources (line 437 and lines 482-485).

Response 8: The references no. 9, and no. 10 have been added in order to justify the recycling rate of 30%:

Line 437: The assumption is made based on general statistical information on plastic waste recycling rates, which is known that only 30% of plastic packaging is being recycled. [9,10].

Line 482: The assumption of disposable single-use plastic cups recycling rate (30%) was made according to the currently available statistical data [9,10].

Line 485: Disposable cups recycling rate (30%) was chosen according to some data provided in the Introduction section [9,10].

Point 9: Review the language.

Response 9: The language has been revised, and changes were marked in a Manuscript.

Reviewer 3 Report

A major improvement to the first version. The introduction adds a lot of value to the paper. Conclusions are clearer and present the findings of the research.

Some tables with final results could help in getting an overview of the results. This can add value to paragraph 3.4 Overview and discussion. Because as you state the participants are of importance to reach final targets. Then the overview is complete. Please consider to use table 1 with the scenario's, type of festival (summarized) and results (percentages).

Textual details:

Line 140: "... co-called". Must be "so-called"?

Line 525: "... on linear waste management model." Should be: a linear waste ..., or: ... linear waste management models.

Line 532: "... polyethylene lining or as a barrier or ...". First 'or' has to be taken out?

Line 547: "... no clear ant comparative ...". ant = and?

Author Response

Point 1: Some tables with final results could help in getting an overview of the results. This can add value to paragraph 3.4 Overview and discussion. Because as you state the participants are of importance to reach final targets. Then the overview is complete. Please consider to use table 1 with the scenario's, type of festival (summarized) and results (percentages).

Response 1: Changes have been made:

The results are summarized in Table 2.

Table 2. Summary overview of the results.

Festival

1

2

3

4

5

6

7

Summary of festival type

High standards for attendance, higher price tickets.

Mature attendees, many foreigners. Strict entrance and ticketing policy.

An open-air event at the city center, free of charge, no restricted territory.

Small punk festival. Attendees are younger age, lower entrance fee.

Electronic music festival. Attendees are younger age, the higher price tickets, far away from the city.

Electronic and another music festival, far away from the city.

Metal, hard rock music.

International, many attendees attend it for many years.

The applied reuse model for beverage cups

A1

A2

B1

B2

B2

B2

C

Only reusable cups, non-refundable model:

one-time eco-fee with a possibility for unlimited exchange of cups

Only reusable cups, with deposit-refund

A mixed system of reusable cups with deposit–refund and of single-use cups

Type of refund

No extra fee for a lost cup

Extra fee for a lost cup

Fully refundable

Partly refundable

Partly refundable

Number of cups per 100 participants per day

133.33

48.66

24.51

54.02

36.08

Share of reusable cups, %

For washing and reuse

78

77.2

96.9

84.9

84.5

86

93.4

85

Damaged, went to recycling

13.5

15.1

1.3

8.6

10.1

5.2

2.5

8.5

Lost

8.5

7.8

1.8

6.5

5.5

7.9

4.1

6.5

Point 2: Line 140: "... co-called". Must be "so-called"?

Response 2: Changes have been made.

Point 3: Line 525: "... on linear waste management model." Should be: a linear waste ..., or: ... linear waste management models.

Response 3: Changes have been made: <…>” linear waste management models”.

Point 4: Line 532: "... polyethylene lining or as a barrier or ...". First 'or' has to be taken out?

Response 4: Changes have been made: “or” was removed from the sentence.

Point 5: Line 547: "... no clear ant comparative ...". ant = and?

Response 5: Changes have been made: “ant” changed to “and”.